



# Impact of horizontal resolution and model time step on European precipitation extremes in the OpenIFS 43r3 atmosphere model

Yingxue Liu[1,2], Joakim Kjellsson[1,2], Abhishek Savita[1] and Wonsun Park[3,4]

[1]GEOMAR Helmholtz Centre for Ocean Research Kiel, Kiel, Germany

[2]Faculty of Mathematics and Natural Sciences, Christian Albrechts University of Kiel, Kiel, Germany

[3]Center for Climate Physics, Institute for Basic Science (IBS), Busan, Republic of Korea

[4]Department of Climate System, Pusan National University, Busan, Republic of Korea

*Correspondence to*: Yingxue Liu (yiliu@geomar.de)

**Abstract**: Events of extreme precipitation pose a hazard to many parts of Europe but are typically not well represented in climate models. Here, we evaluate daily extreme precipitation over Europe during 1982–2019 in observations (GPCC), reanalysis (ERA5) and a set of atmosphere-only simulations at low- (100 km), medium- (50 km) and high- (25 km) horizontal resolution with identical vertical resolutions using OpenIFS (version 43r3). We find that both OpenIFS simulations and reanalysis underestimate the rates of extreme precipitation compared to observations. The biases are largest for the lowest resolution (100 km) and decrease with increasing horizontal resolution (50 and 25 km) simulations in all seasons. The sensitivity to horizontal resolution is particularly high in mountain regions (such as the Alps, Scandinavia, Iberian Peninsula), likely linked to the sensitivity of vertical velocity to the representation of topography. The sensitivity of precipitation to model resolution increases dramatically with increasing percentiles, with modest biases in the $70^{th}$–$80^{th}$ percentile range and large biases above the $99^{th}$ percentile range. We also find that precipitation above the $99^{th}$ percentile mostly consists of large-scale precipitation (~80 %) in winter, while in summer it is mostly large-scale precipitation in Northern Europe (~70 %) and convective precipitation in Southern Europe (~70 %). Compared to ERA5, the OpenIFS overestimates large-scale precipitation extremes in winter, but underestimates in summer. The discrepancy between OpenIFS and ERA5 decreases with increasing horizontal resolutions. We also examine the sensitivity of extreme precipitation to model time step and find that the convective contribution to extreme precipitation is more sensitive to the model time step than the horizontal resolution. This is likely due to the sensitivity of convective activity to model time step. On the other hand, the large-scale contribution to extreme precipitation is more sensitive to horizontal resolution than the model



time step, which may be due to sharper fronts and steeper topography at higher horizontal
resolution.

## 1. Introduction


Extreme precipitation events have severe impacts on human society and ecosystems. For
example, Germany experienced extreme precipitation during mid-July 2021, which exceeded
100 mm/d over a large area resulting in a devastating flood. The recent flood is one of the most
serious natural disasters for Germany since 1962, in which around 180 people died as a result
of the flood. Coupled Model Intercomparison Project (CMIP) models are used to understand
the present and future climate. The CMIP5 models project that the frequency of the most
intense precipitation observed today in Europe would be almost double in the future at each
1°C of warming (Myhre et al., 2019). Recently the CMIP6 models also projected an increase
in extreme precipitation over most of the regions under global warming (Intergovernmental
Panel on Climate Change, 2023; Li et al., 2021). The increasing extreme precipitation poses a
threat for society and must thus be realistically simulated and projected accurately for future
climates. However, the climate models have large uncertainties in simulating extreme
precipitation events due to lack of observations, coarse horizontal resolution grid, long model
time step etc. (Alexander et al., 2019; Avila et al., 2015; Sillmann et al., 2013). This study aims
to understand the sensitivity of extreme precipitation to model resolution and time step.
The CMIP models can simulate time-mean precipitation very well but usually underestimate
the intensity and frequency of extreme precipitation (O'Gorman, 2015; Sillmann et al., 2013).
The intensity of simulated extreme precipitation often increases with increased horizontal
resolution in atmosphere models (Caldwell, 2010; Rauscher et al., 2016; Wehner et al., 2010,
2014). Jong et al. (2023) analyzed the extreme precipitation in Northeastern United States (US)
using the Seamless system for Prediction and EArth system Research (SPEAR) and Large
Ensemble of Community Earth System Model version 1 (CESM-LE) model simulations at
different horizontal resolutions, and they found that a model with 25 km horizontal resolution
simulates much more realistic frequency, amplitude, temporal variability and trends in extreme
precipitation than 50 and 100 km model resolution. However, Kopparla et al. (2013) found that
the reduced biases at higher horizontal resolution do not hold for all regions. They concluded
that extreme precipitation with finer model resolutions in Community Atmospheric Model





version 4 (CAM4) has better agreement with observational datasets in Europe and the US, but
not in Australia.
Considering the time and computational cost, climate simulations of more than 100 years are
generally not feasible with high-resolution (25 km or higher) models. Instead, regional climate
models (RCMs) are developed by focusing on a particular region, where higher-resolution
model simulations can be conducted with reduced cost (Laprise, 2008). Strandberg & Lind
(2021) compared the precipitation using both global (CMIP5, CMIP6 and HighResMIP) and
regional (CORDEX RCMs) model simulations at different resolutions (~300−12.5 km) and
found that high-resolution models reduce the biases for extreme precipitation. They also found
that the effect of horizontal resolutions for extreme precipitation is mostly in regions with
complex topography and in the summer season when precipitation is mostly caused by
convective processes, in agreement with Iles et al. (2020). The reduced biases in extreme
precipitation near topography in high-resolution models is mostly due to an improved
representation of topography, coastlines, and small-scale processes such as convection and
diffusion. However, Strandberg & Lind (2021) showed that models with higher horizontal
resolution overestimate the intensity of extreme precipitation in some regions over Europe.
Moreover, once reaching 50 km, the improvement is smaller for further higher resolution,
which is consistent with Demory et al. (2020), as they found the effect of increasing resolution
from 50 to 12 km grid on the daily precipitation distributions is smaller outside the mountainous
and coastal regions. However, Chan et al. (2013) investigated the precipitation in regional
models with 50, 12 and 1.5 km grid spacing over the southern UK and found that the
representation of daily orographic precipitation improved when increasing horizontal
resolution from 50 km to 12 km, but not from 12 km to 1.5 km. Chan et al. (2013) found that
1.5 km simulations (convection-permitting) predominantly improve the representation of
extreme precipitation on sub-daily timescales but not for daily timescales, which is further
consistent with Prein et al. (2013). The small improvements for extreme precipitation in higher
horizontal resolution simulations indicate that although the bias of daily extreme precipitation
is reduced with finer horizontal resolution, there is also a "diminishing return".
No global atmosphere-model simulations in the Atmosphere Model Intercomparison Project
(AMIP) in CMIP6 explicitly resolve convection and all must therefore employ
parametrizations of such motions and users must carefully choose the associated parameters.
The cloud microphysics is sensitive to the model time step in an idealized convection-



permitting model, e.g., the precipitation is reduced 53 % when the time step was lengthened from 1s to 15 s (Barrett et al., 2019). Mishra & Sahany (2011) also found a more realistic simulation of the precipitation pattern in the tropics when the time step was shortened from 60 min to 5 min. Wan et al. (2021) found that 10-year mean zonal averages change when the time step is reduced by a factor of 6, such as the temperature, cloud fraction, and the relative humidity in the troposphere. Bador et al. (2020) showed that models at higher horizontal versions (50 km or 25 km) where convection parameters were not re-tuned to the increased resolution often exhibit larger biases than corresponding model versions at lower horizontal resolution.

A recent study by Savita et al. (2024) explored the sensitivity of global mean precipitation to the horizontal resolution and model time step in atmosphere-only simulations with OpenIFS. However, the extreme precipitation sensitivity to horizontal resolution and time step was not investigated. In this study, we investigate the impact of horizontal resolutions (~100 km, ~50 km, and ~25 km) and model time steps (60 minutes, 30 minutes, and 15 minutes) on daily extreme precipitation using OpenIFS simulations and compare them with observation. This paper is structured as follows: section 2 describes the data and methodology, and section 3 discusses the results. The conclusion and discussion can be found in section 4.

## 2. Data and Methods

### 2.1 Model, observation, and reanalysis data

The OpenIFS is derived from the Integrated Forecasting System at the European Centre for Medium-range Weather Forecasting (ECMWF-IFS) cycle 43 release 3 (43r3) (ECMWF, 2017). We use the same AMIP simulations that were used in Savita et al. (2024) which cover the period 1979-2014 and are extended to 2019 using sea-surface temperature (SST) from ERA5 and the Shared Socioeconomic Pathway 5 (SSP5-8.5) scenario from CMIP6. OpenIFS simulations use 91 vertical levels (L91) and the different horizontal resolutions: low resolution (Tco95, ~100 km), medium resolution (Tco199, ~50 km), and high resolution (Tco399, ~25 km). For the low resolution, additional sensitivity experiments use different model time steps i.e., 60, 30, and 15 minutes and we refer to these experiments as LR60m, LR30m, and LR, respectively. For medium and high resolution, the same model time step is used (i.e., 15 minutes), of which experiments refer to as MR and HR, respectively. While the OpenIFS uses





a reduced octahedral grid (Malardel et al., 2016), the final output used in this study has been
interpolated to a regular grid using the XIOS output server. The LR, LR30m and LR60m data
were interpolated to a global 0.9° grid while the MR and HR data were interpolated to a global
0.45° grid, i.e., we are not investigating extreme precipitation in high resolution simulations in
their native grid, which will be investigated in future study. The simulations used here were
used by Savita et al. (2024) who found improvements in the surface zonal wind, Rossby wave
amplitude and phase speed, weather regime patterns, and surface-air temperature when
reducing a model time step from 60 minutes to 30 and 15 minutes in low resolution or
increasing the horizontal resolution from 100 km to 50 and 25 km. However, Savita et al. (2024)
did not find such improvement in the mean precipitation bias by increasing horizontal
resolution or reducing the model time step.
To validate OpenIFS simulations, we use the gridded daily precipitation observational data
from Global Precipitation Climatology Centre (GPCC) with resolution of $1° \times 1°$ for the period
1982–2019 (Ziese et al., 2022) as well as the reanalysis data from the ECMWF Reanalysis v5
(ERA5) for 1979–2019 (Hersbach et al., 2023). ERA5 is based on the IFS Cy41r2, with 31 km
horizontal resolution and 137 levels (Hersbach et al., 2020). We analyzed total, large-scale,
and convective precipitation in this study. The total precipitation (convective plus large-scale
precipitation) in the IFS is the accumulated precipitation, comprising of rain and snow, that
falls to the Earth's surface, and it is not assimilated in the IFS. The convective precipitation is
generated by the convection scheme in the IFS, which represents convection at spatial scales
smaller than the grid box. The convection scheme follows Sundqvist (1978), which is also used
in the OpenIFS. The large-scale precipitation is generated by the cloud scheme (Khairoutdinov
& Kogan, 2000), which represents the formation and dissipation of clouds and large-scale
precipitation due to changes in atmospheric quantities (such as pressure, temperature, and
moisture) predicted directly by the IFS at spatial scales of the grid box or larger. The
autoconversion/accretion parameterization is a non-linear function of the mass of both liquid
cloud and rainwater. The calculation follows Khairoutdinov & Kogan (2000) which is derived
from large eddy simulation studies of drizzling stratocumulus clouds, and this scheme is also
used in OpenIFS. Several studies have evaluated the performance of ERA5 and found that the
total precipitation in ERA5 is performing well over the US (Tarek et al., 2020; Xu et al., 2019).
For global precipitation, the mean absolute difference over 50° S–50° N between ERA5 and
TRMM/3B43 is 0.58 mm/d; the global-mean correlation with GPCP data is 0.77, which is





better compared to ERA-Interim (0.63 mm/d and 0.67) (Hersbach et al., 2020). ERA5 also
performs well in polar regions in representing wind, temperature and humidity (Graham et al.,
2019; Tetzner et al., 2019; Wang et al., 2019).
Here we analyze daily ERA5 and the OpenIFS data over Europe (30° N–72° N, 10° W–40° E)
for the period of 1982–2019 to be consistent with GPCC dataset. For comparison, the ERA5,
GPCC, MR, and HR data are remapped to LR (~0.9375° × 0.9375°) using the second-order
conservative remapping method. The second-order conservative method includes the gradient
across the source cell, which is not included in the first-order conservative method. Therefore,
it gives a smoother, more accurate representation of the source field (Jones, 1998).

**2.2 Methods**
**Calculation of $q^{th}$ percentile value**
We calculated different percentile values using total precipitation from GPCC, ERA5, and
OpenIFS simulations. When we calculated the $q^{th}$ percentile value, the normalized ranking
usually did not match the location of the $q^{th}$ percentile exactly, which means the $q^{th}$ lies between
two indices. Therefore, we determined the location first, then computed the $q^{th}$ value by
interpolating between the two nearest values based on the location. Here we used the formula
below to find the location:

183                          $j = q*(n-1)$                          (1)

$n$ is the length of the sample, $q$ is the desired percentile, $j$ is the location which is the distance
from the first value $X_1$ ($X_m$ are the sorted sample values, $m$=1, 2, …, $n$). Then we took $i$ as the
nearest (lower) integer of $j$ to get the $q^{th}$ value P(q) by interpolating.

187                     $P(q) = X_i + (X_{i+1} - X_i) * (j-i)$                     (2)

There are other methods to determine the location of $q^{th}$ percentile (Hyndman & Fan, 1996),
but here we use the 'linear' one.

**The convective contribution to extreme precipitation**
To calculate the contribution of convective precipitation to total precipitation for a percentile
range, at each grid point we accumulated the convective precipitation on all days when the total
precipitation is in that percentile range, then divided it by the accumulated total precipitation
on those days to get the fraction of convective precipitation.






**Calculation of RMSE values**
We used the root-mean-square error (RMSE) referenced to GPCC that measures the
performance of ERA5 and OpenIFS simulations:

$$\text{RMSE} = \sqrt{\frac{\sum_{i=1}^{n}(x_{mi}-x_{oi})^2}{n}} \qquad (3)$$

$x_{mi}$ is the value at $i$ grid point for ERA5 or OpenIFS simulations, $x_{oi}$ is the value for GPCC, $n$
is the number of land grid points over Europe. Using equation (3), we calculated the RMSE
values for different percentile ranges. Smaller RMSE values mean the biases between OpenIFS
(or ERA5) and GPCC are smaller i.e., the model simulations and ERA5 are performing better.

**Confidence intervals**
We calculated the 2.5 to 97.5[th] confidence intervals (CI) for the RMSE for each percentile with
a bootstrap method. For example, to calculate the CI for the RMSE of HR (referenced to GPCC
observation), we randomly chose $n$ grid cell pairs from GPCC and HR over European land,
then calculated their RMSE ($n$ is the number of total land grid points over Europe). This process
was repeated for *2000* times. We took the 2.5[th] and 97.5[th] percentiles of the distribution of the
2000 RMSEs as the 95 % CI. If the CI for different simulations do not overlap then we refer
that they are significantly different.

**3. Results**

**3.1 Extreme precipitation over Europe**

We show the time series of 99[th] percentile precipitation calculated from all grid points and all
days in each year over the period 1982–2019 from GPCC, ERA5, and OpenIFS simulations
over Europe (Fig. 1). The ERA5 simulates an inter-annual variability of the 99[th] percentile
precipitation similar to that in GPCC. For example, the peak in 2010 and the low in 1994 are
well reproduced in the ERA5. OpenIFS simulations do not reproduce the same inter-annual
variability as in GPCC or ERA5 but LR and HR do reproduce the 95 % significant positive
trend observed in GPCC (0.03 mm/d/y, not shown), which are ~0.2 mm/d/y for both LR and
HR, and it is not significant for MR. We note that the OpenIFS simulations use observed SST
and sea-ice concentrations as boundary conditions, but ozone is taken from a photochemical
equilibrium (Cariolle & Teyssèdre, 2007) and aerosol concentrations are taken from
Copernicus Atmosphere Monitoring Service (CAMS) monthly climatology. Therefore, we do
not expect LR, MR and HR to reproduce trends driven by ozone or aerosols forcing. We also





find that both ERA5 and OpenIFS simulations have relatively lower 99th percentile
precipitation rates compared to GPCC (Fig. 1). The RMSE for ERA5 (0.36 mm/d) is lower
than for OpenIFS simulations which is largest for LR (2.03 mm/d) and decreases with
increasing horizontal resolution (i.e., 1.13 mm/d for MR and 0.69 mm/d for HR). Note that Fig.
1 does not contain any spatial information and that a mismatch between model data and
observations can be due to the 99th percentile occurring in different regions and/or with
different magnitudes. The RMSE analysis suggests that ERA5 and HR are close to GPCC and
LR is far from GPCC.

Figure 2a–e shows the spatial distribution of the 99th percentile precipitation over Europe for
all days in each season for all years in GPCC, ERA5, and OpenIFS simulations, respectively.
In general, the extreme precipitation is very low (~ 2 mm/d) in Northern Africa, which is to be
expected since the mean precipitation is only 0.5 mm/d in those regions (Fig. S1). The extreme
precipitation exceeds 30 mm/d over mountain areas (e.g., Scandinavian mountains, Alps, and
Iberian Peninsula) and the north coast of the Mediterranean but is otherwise lower (~15 mm/d).
The spatial distribution of extreme precipitation matches that of the mean precipitation pattern
(Fig. S1). The high 99th percentile precipitation near mountains is likely due to the forced ascent
of westerly (Scandinavia, Iberian Peninsula, British Isles) and southerly (Alps) winds. The high
99th precipitation in the north of the Mediterranean is likely because of warm and moist
southerly winds from the Mediterranean Sea. The ERA5 and OpenIFS simulations overall
reproduce the spatial distribution of the 99th percentile precipitation from GPCC. However, the
magnitudes are different, particularly over the Scandinavian mountains, the Alps, and central
Europe near 50° N (Fig. 2a–e). Figure 2f–i show the regional biases for the 99th percentile
precipitation referenced to GPCC. LR mostly underestimates the 99th percentile precipitation
in mountainous areas and deserts by more than 25 % (Fig. 2g) and the biases are reduced when
horizontal resolution is increased in MR and HR (Fig. 2h–i). We also notice that LR
underestimates the 99th percentile precipitation south of the Alps but overestimates it to the
north (Fig. 2 (g)), whereas this bias is negligible in higher-resolution simulations (Fig. 2h–i).
Lavers et al. (2022) also found too much extreme precipitation on the north side of the Alps in
ERA5 during a storm. This could be because the moist southerly winds do not ascend high
enough with LR, therefore there is less precipitation formed on the southern side and more
moisture is advected over the mountain. The reduced biases near mountain regions in the
higher-resolution simulations are likely because higher resolution has a better representation



of topography and vertical velocity. A cross section of the topography and annual-mean vertical
velocity at 850 hPa and 62° N (Fig. S2 and S3) highlight that the higher-resolution simulations
resolve steeper topography, which leads to more ascent and thus more precipitation.

The $99^{th}$ percentile precipitation over the Alps is more realistic with higher horizontal
resolution compared to lower resolution. However, all simulations as well as ERA5 exhibit a
negative bias over northeast Italy and west Slovenia (Fig. 2f–i). The cause could be a bias in
GPCC or a persistent model bias in the ECMWF-IFS on which both ERA5 and OpenIFS are
based. In general, ERA5 has a lower RMSE (2.6 mm/d) for extreme total precipitation than
OpenIFS simulations, i.e., ERA5 has overall lower biases than LR (4.0 mm/d) and is similar to
MR (3.0 mm/d) and HR (2.9 mm/d).

We next calculate the trend for the annual $99^{th}$ percentile precipitation over Europe (Fig. S4 &
S5) and find that the $99^{th}$ percentile precipitation has a large positive trend in central Europe
and a negative trend to the north of the Alps in GPCC (Fig. S4a). The ERA5 reproduces the
pattern of the trend found in GPCC but not significant. However, OpenIFS simulations do not
have consistent patterns with GPCC (Fig. S4c–e, Fig. S5c–e), with only LR60m reproducing
the large positive trend in central Europe (Fig. S5c). Overall, the trend is largely underestimated
over central Europe but overestimated over northern Europe in OpenIFS simulations. We have
not found any consistent improvement across the horizontal resolution and model time step.

In addition to the $99^{th}$ percentile precipitation, we calculate annual total precipitation in
different percentile ranges, such as $70^{th}$–$80^{th}$, $80^{th}$–$90^{th}$, $90^{th}$–$95^{th}$, $95^{th}$–$99^{th}$, $99^{th}$–$99.5^{th}$, $99.5^{th}$–
$99.9^{th}$ and larger than $99.9^{th}$ (i.e., $>99.9^{th}$) percentile. We calculate the RMSEs for ERA5 and
OpenIFS simulations referenced to GPCC in each range and find that the RMSEs for ERA5
and OpenIFS simulations vary strongly across percentile ranges (Fig. 3). The RMSEs increase
exponentially with increasing percentiles, from less than 1 mm/d at the $70^{th}$–$80^{th}$ percentile
range to ~8 mm/d above the $99.9^{th}$ percentile range. The largest RMSE is found for LR60m
above the $99.9^{th}$ percentile range which is around 12 mm/d [CI: 11.3–12.8 mm/d]. We also find
that the RMSEs decrease with finer horizontal resolution for all percentile ranges. The CI of
the RMSEs from LR do not overlap with those from higher horizontal resolutions for any
percentile range, i.e., the biases from LR are significantly different from that at higher
resolutions and thus clearly sensitive to the horizontal resolution. We also find that the RMSE



differences between LR simulation and the higher-resolution simulations as well as ERA5 are
larger at higher percentile ranges (>95th) than those at lower percentile ranges (<95th). Thus,
we conclude that extreme precipitation is more sensitive to horizontal resolution than
precipitation at lower percentile ranges (<95th). ERA5 has the smallest RMSE of all datasets
above the 95th percentile ranges, i.e., ERA5 has a better representation of the extreme
precipitation than our OpenIFS simulations (Fig 3).

The RMSEs for LR60m, LR30m, and LR are increasing with increasing model time steps.
However, the CI of RMSE overlap at all percentile ranges, i.e., the sensitivity of precipitation
to the model time step is not statistically significant in the low-resolution configurations. While
the model time step may influence precipitation, especially convective precipitation, errors
from poorly resolved topography probably have a large impact on the RMSE, which would
explain the lack of sensitivity to the model time step.

**312    3.2 Relative roles of convective and large-scale precipitation**
We calculate the fractions of convective and large-scale precipitation in total precipitation for
days when the total precipitation exceeds the 99th percentile in all model simulations and ERA5
(Fig. 4 & 5). Note that, GPCC does not provide convective and large-scale precipitation
separately, therefore we compare our OpenIFS simulations to the ERA5 dataset to assess the
realism of the model simulations. We note that ERA5 is a reanalysis dataset where precipitation
is a parametrized variable, and observations of which are not assimilated over Europe. ERA5
monthly precipitation has a good agreement with GPCC on the land, with correlations above
90 % for most of Europe, and above 70 % for Australia, Asia, and North America (Bell et al.,
2021). ERA5 also shows smaller biases for mean precipitation than other reanalysis datasets in
the tropics compared to the Global Precipitation Climatology Project (GPCP), with relative
biases of 13 % for ERA5, 17 % for MERRA-2 and 36 % for JRA-55 (Hassler & Lauer, 2021).
The biases for mean precipitation are found smaller over extra-tropics than the tropics
compared to the gauge-based precipitation observations, particularly agreeing well with
observations over central Europe and South Asia (Lavers et al., 2022). Moreover, ERA5 can
capture the locations and patterns of highest precipitations in observations, but cannot simulate
the magnitude (Lavers et al., 2022). We also find that the extreme precipitation over Europe in
ERA5 is closer to observations than models (Fig. 1, 2, and 3), therefore, we use ERA5 for the
benching mark here although it has some known biases.

The ERA5 data and OpenIFS simulations show that, in DJF, the extreme precipitation is nearly
100 % large-scale precipitation over northern Europe, more than 90 % over central Europe,
and more than 70 % over western and southern Europe (Fig. 5a–d). However, in JJA, large-
scale precipitation makes up most of the extreme precipitation over northern Europe (>70 %)
while convective precipitation makes up most of the extreme precipitation in the Mediterranean
region (>70 %) (Fig. 4a–d). The OpenIFS simulations largely reproduce the pattern of the
fraction of convective precipitation found in ERA5, but we note differences in magnitudes (Fig.
4e–g, and Fig. 5e–g)). In JJA, the OpenIFS simulates the contribution of the convective
precipitation quite well over Scandinavia where the extreme precipitation is mostly large-scale
precipitation, but overestimates that for other areas over Europe (Fig. 4e–g). The RMSEs from
MR (0.10 mm/d [CI: 0.09–0.10 mm/d]) and HR (~0.09 mm/d [CI: 0.09–0.10 mm/d]) are not
significantly different, while LR (~0.12 mm/d [CI: 0.12–0.13 mm/d]) is significantly larger
than those in MR and HR. In DJF, the OpenIFS underestimates the contribution from
convective precipitation except for near-coastal areas (Fig. 5e–f). That is, the contribution from
large-scale precipitation is overestimated, and the bias is reduced with higher horizontal
resolution, i.e., in MR and HR.

Further, we explore the relative roles of horizontal resolution and time step for the large-scale
and convective precipitation at different percentile ranges (Fig. 6). In general, the RMSEs
increase with increasing percentiles, but also decrease with increasing horizontal resolution and
shorter model time step. The RMSE reduces for higher percentile in higher resolution due to
better representation of topography, and in smaller model time step due to enhanced convection.
The exceptions are the total precipitation above the 99.5th percentile in JJA where the RMSEs
from LR are larger than LR30m (Fig. 6a), and the convective precipitation above the 99th
percentile in JJA and DJF where the RMSEs from HR are larger than MR (Fig. 6c & f).

The CI for RMSEs of total precipitation from LR, MR and HR in DJF and JJA do not overlap
for all percentile ranges, thus there is a significant sensitivity of the total precipitation to the
horizontal resolution, particularly for extreme total precipitation. The exceptions are the total
precipitation below the 90th percentile ranges and above the 99.9th percentile range in JJA
where the CI for RMSEs in MR and HR overlap (Fig. 6a). However, the sensitivity is not found
for the global-mean total precipitation by increasing horizontal resolution (Savita et al., 2024).



For the large-scale precipitation in JJA, the CI for RMSEs from LR do not overlap with those
from MR and HR at higher percentile ranges (>95th), but overlap at lower percentile ranges
(<95th) (Fig. 6b). That is, the large-scale precipitation from the extreme precipitation is
sensitive to the horizontal resolution. We note that a reduced bias is not found for the
convective precipitation in JJA (Fig. 6c), and we conclude that the horizontal resolution
dependence of extreme total precipitation in JJA comes from the large-scale precipitation. For
DJF, the large-scale precipitation is sensitive to the horizontal resolution for all percentile
ranges, where the CI for RMSEs in LR do not overlap with those from MR and HR (Fig. 6e).
The convective precipitation in DJF is also sensitive to the horizontal resolution (Fig. 6f),
however there is little convection precipitation in DJF, thus the sensitivity for convective
precipitation in DJF is not important. Therefore, the resolution dependence of extreme total
precipitation is mostly dominated by the large-scale precipitation in DJF.

For the model time step, the CI for RMSEs of total precipitation from LR60m, LR30m, and
LR overlap at all percentile ranges in both JJA and DJF (Fig. 6a & d), i.e., the extreme total
precipitation is not sensitive to the model time step in a significant way in the low-resolution
simulations. Similarly, the mean total precipitation is also found insensitive to the model time
step (Savita et al., 2024). Both the large-scale and convective precipitation are sensitive to the
model time step particularly above the 95th percentile ranges in JJA (Fig. 6b & c). The
convective precipitation is more sensitive to the model time step than the large-scale
precipitation in JJA, but in DJF the sensitivity is found only for the large-scale precipitation
(Fig. 6e). Also, the lack of sensitivity for convective precipitation in DJF may be because there
is almost no convective precipitation in DJF.

**4. Discussion and Conclusion**
We have investigated the sensitivity of extreme precipitation across different horizontal
resolutions and model time steps in atmosphere-only experiments with the OpenIFS.
Comparing extreme precipitation (defined as total daily precipitation at the 99th percentile)
from OpenIFS simulations, reanalysis (ERA5), and observation (GPCC), we find that MR and
HR mostly better represent the precipitation extremes compared to LR. We also found a more
significant sensitivity to the horizontal resolution for the precipitation above the 95th percentile
and less sensitivity for lower percentile ranges (<95th) (Fig. 3). These OpenIFS-based results
are similar to Kopparla et al. (2013), who found that the bias of extreme precipitation in the
high-resolution simulation (25 km) is reduced compared to the lower-resolution simulations





(100 km and 200 km) over Europe in their atmospheric model, but not for precipitation at lower
percentiles (i.e., <95[th]). However, the sensitivity to the horizontal resolution found by Kopparla
et al. (2013) was not significant over Europe which is rather different from our results as we
have found a significant difference across the horizontal resolutions. In contrast to the extreme
precipitation, the bias for global mean precipitation is not decreasing when increasing
horizontal resolution from ~200 km to ~100 km or ~50 km in the ECHAM6-AMIP simulations
(Hertwig et al., 2015), and also in other GCMs (e.g., OpenIFS, HadGEM1 and HadGEM3)
(Demory et al., 2020; Savita et al., 2024; Schiemann et al., 2014). However, Delworth et al.
(2012) found an improvement in the global mean precipitation with increasing horizontal
resolution in a coupled model (GFDL).

The improvements due to increasing horizontal resolution for the extreme precipitation are
mostly over the mountain areas, consistent with previous studies which found the effect of
horizontal resolution being largest in areas with complex topography over Europe and also
other regions for mean and extreme precipitation (Demory et al., 2020; Iles et al., 2020;
Monerie et al., 2020; Prein et al., 2013; Torma et al., 2015). The sensitivity to the horizontal
resolution comes from the large-scale precipitation, which is likely because of the better-
resolved topography. However, the convective precipitation in JJA is more sensitive to the
model time step than it is to the horizontal resolution, likely because there is an increase in
shallow and mid-level convection with a shorter time step in the OpenIFS (Savita et al., 2024),
thus we get more convective precipitation.

In our results, larger improvements are obtained when the horizontal resolution is increased
from LR to MR, but relatively smaller improvements from MR to HR. Similar results are also
found in Roberts et al. (2018), where the climatological surface biases are relatively insensitive
when increasing the atmospheric resolution from ~50 km to ~25 km in the ECMWF-IFS. Jung
et al. (2012) also showed that the largest improvements in extratropical cyclones, Euro-Atlantic
blocking, tropical mean precipitation, and tropospheric circulation are found when increasing
horizontal resolution from 126 km to 39 km with relatively small further changes from 39 km
to 16 and to 10 km in ECMWF atmosphere model. Kopparla et al. (2013) and Bacmeister et al.
(2014) found much improvement for the mean precipitation and extreme precipitation with the
increasing atmospheric resolution from ~200 km to 100 km, but less improvement from ~100
km to ~25 km. It is likely due to a lack of tuning with the changing horizontal resolution. The
above conclusions are valid over Europe, but also valid for other regions such as the tropics



and subtropics. For example, the predictions of tropical cyclone intensity are markedly
improved when the horizontal resolution of the atmosphere model is increased from 120 km to
40 km, but not for 15 to 10 km (Jung et al., 2012), which often triggers extreme precipitation
(Gori et al., 2022; Zhu & Quiring, 2022).

Moreover, the choice of observation dataset is a key factor for assessing the impact of the
horizontal resolution and model time step on extreme precipitation. Most observation
precipitation data are from one of the three categories, gauge-based products, satellite products,
and merged satellite-gauge products. Since the satellite products are constructed with satellite
microwave and/ or infrared measurements, with/ without gauged-adjusted estimates,
differences exist between these products. Besides, the gauge-based products are highly
dependent on the choice of stations and interpolation schemes. It is hard to say which product
is closer to reality, as different regions may have different observation datasets that suit best
for the analysis. In particular, we note that not all products are suitable for extreme analysis.
For example, GPCP's main scope is to construct a reliable climate data record and has been
developed with a priority of ensuring the long-term stability of data (Adler et al., 2017).
Masunaga et al. (2019) found that the frequency of GPCP daily precipitation quickly drops
below all other datasets once the precipitation exceeds 30 mm/d. Also, the time series of GPCP
extreme precipitation over the ocean exhibits a jump to lower $99^{th}$ percentiles in late 2008/early
2009 which is not present in all other datasets, coinciding with the change in utilization of
SSM/I and SSMIS. The lower $99^{th}$ precipitation suggests that the GPCP dataset might not be
applied to extreme analysis (Masunaga et al., 2019). Therefore, we only use GPCC observation
data as the reference to explore the model performance. In Fig. 2f–i the $99^{th}$ percentile
precipitation is largely underestimated in the eastern Alp region by ERA5 and all model
simulations. The biases are insensitive to horizontal resolution. It is likely a persistent model
bias in the ECMWF-IFS or a bias in GPCC. Analyzing multiple precipitation products instead
of relying on a single one may be a good way to reduce these biases.

**Code and data variability**
The OpenIFS model requires a software license agreement with ECMWF to use it, and
OpenIFS's license is easily given free of charge to any academic or research institute. The
details          of          OpenIFS          are          available          at
https://confluence.ecmwf.int/display/OIFS/About+OpenIFS (ECMWF, 2018). We used the
same simulation that used in Savita et al. (2024) and therefore do not provide the data needed



to reproduce the simulations here. All data (runscripts, input data etc) needed to reproduce the
simulations can be found in Savita et al. (2024) in code and data variability section. The jupyter
notebook scripts used in this study to produce the plots can be found at
https://doi.org/10.5281/zenodo.10887652. The raw model output is available from the authors
upon reasonable request. The observation and reanalysis datasets used in this study can be
downloaded from GPCC (https://opendata.dwd.de/climate_environment/GPCC/html/fulldata-
daily_v2022_doi_download.html,      Ziese      et      al.,      2022)      and      ERA5
(https://cds.climate.copernicus.eu/cdsapp#!/dataset/reanalysis-era5-single-levels?tab=form,
Hersbach et al., 2023).

**Authors contributions.**   AS and JK conducted all the OpenIFS simulations. YL did the
analysis and writing with substantial contribution from JK, AS and WP.

**Competing interests.**   The contact author has declared that none of the authors has any
competing interests.

**Acknowledgements.**  Yingxue Liu is supported by China Scholarship Council (CSC, grant no.
202004910401). Joakim Kjellsson and Abhishek Savita are supported by JPI Climate/Ocean
(ROADMAP project, grant no. 01LP2002C). Wonsun Park was supported by IBS (grant no.
IBS-R028-D1). We thank the OpenIFS team at ECMWF for the technical support. All the
OpenIFS simulations were conducted on the HLRN machine under shk00018 project resources.
All the analysis and data storage were conducted on computer clusters at GEOMAR and Kiel
University Computing Center (NESH).

**Financial support.** This research is financially supported by CSC (grant no. 202004910401)
and ROADMAP project (grant no. 01LP2002C).



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





694                                    **Figures**

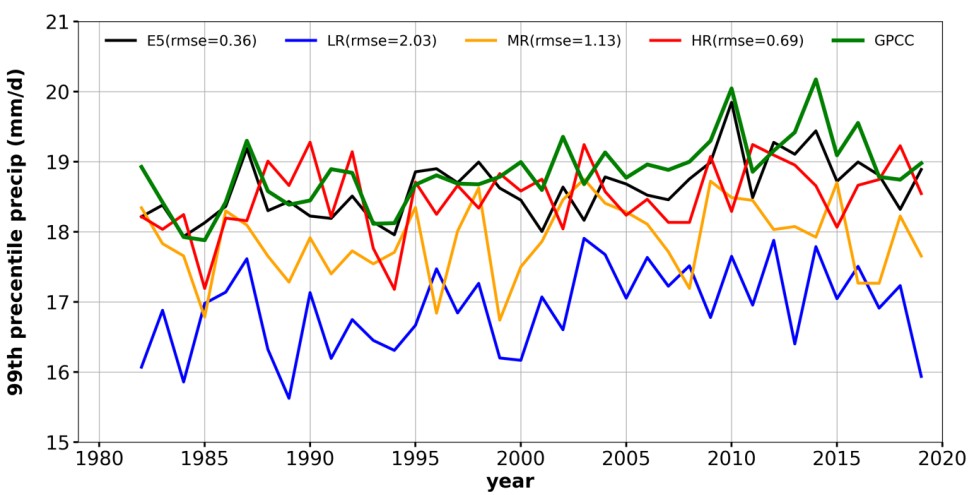


**Fig. 1** Annual time series of the 99[th] percentile precipitation using observations (GPCC, green),
reanalysis (ERA5, black), and model simulations (LR: blue, MR: orange, HR: red) during
1982-2019 over Europe. RMSE values of 99[th] percentile precipitation are computed referenced
to GPCC which are shown within the small bracket.




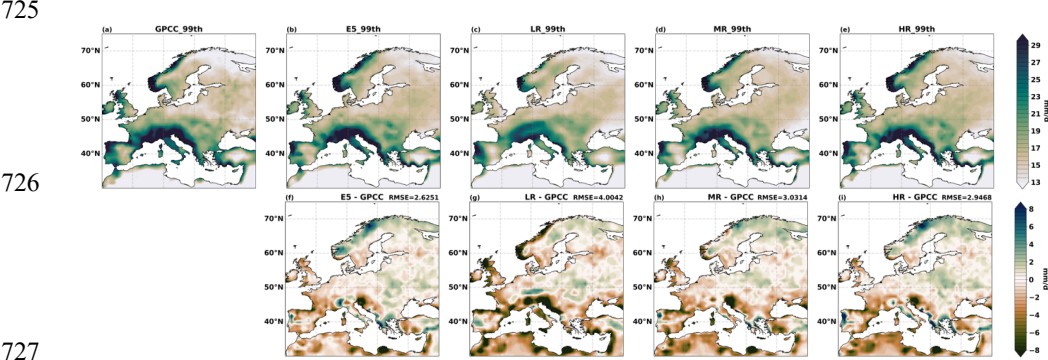


**Fig. 2** The 99th percentile precipitation over Europe during 1982-2019 from (a) GPCC
observations, (b) ERA5 reanalysis, (c) LR, (d) MR, (e) HR, and the corresponding biases and
RMSEs in (f) ERA5, (g) LR, (h) MR, and (i) HR.






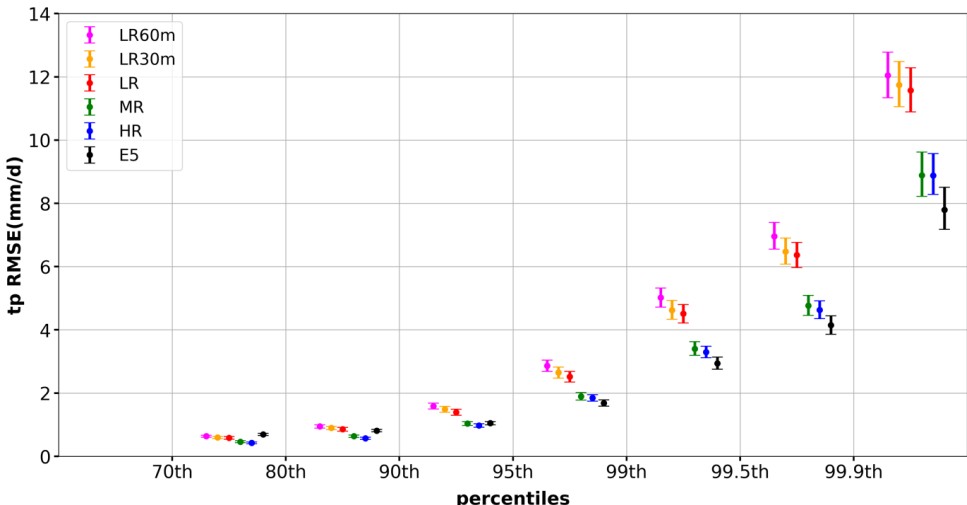


**Fig. 3** RMSEs for annual total precipitation at different percentile ranges (70th – 80th, 80th –

90th, 90th – 95th, 95th – 99th, 99th – 99.5th, 99.5th – 99.9th and >99.9th percentile) in ERA5 (black)

and OpenIFS simulations (LR60m: magenta, LR30m: orange, LR: red, MR: green, HR: blue)

referenced to GPCC during 1982-2019 over Europe. Dots are the RMSE values, and error bars

are the 95 % CI.






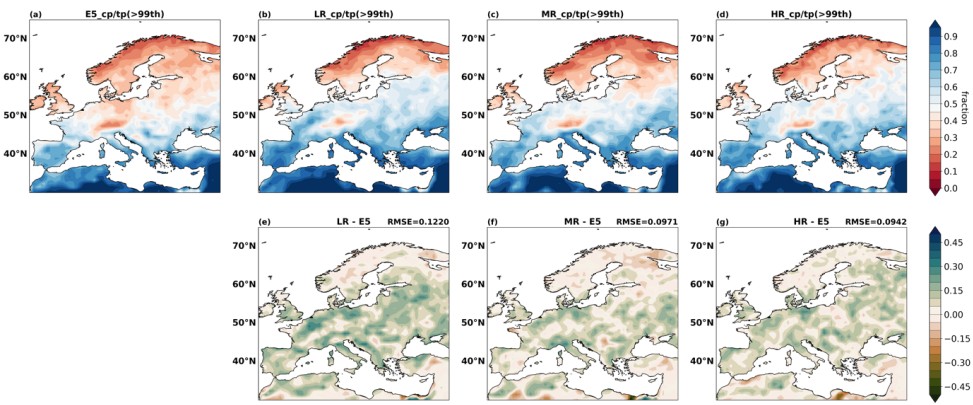

**Fig. 4** Contribution of convective precipitation to extreme precipitation (>99$^{th}$ percentile) in (a) ERA5, (b) LR, (c) MR and (d) HR over Europe in JJA, and (e)– (g) their biases and RMSEs compared to ERA5 over the period 1982-2019.

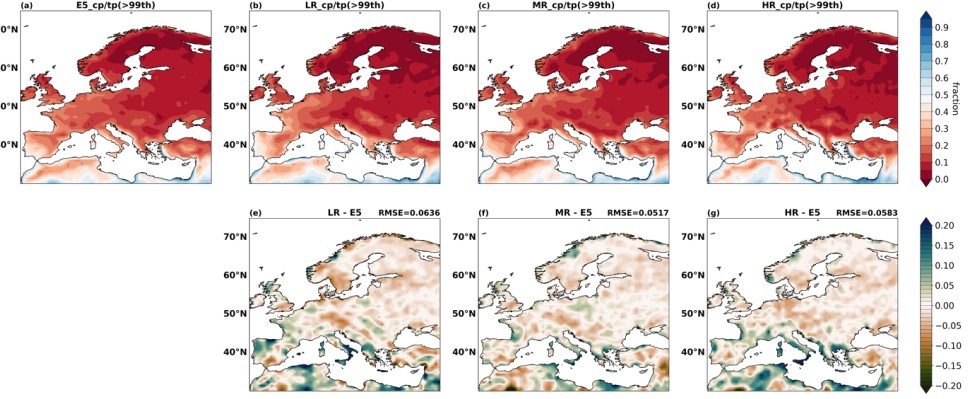

**Fig. 5** The same as Fig. 4 but for DJF.




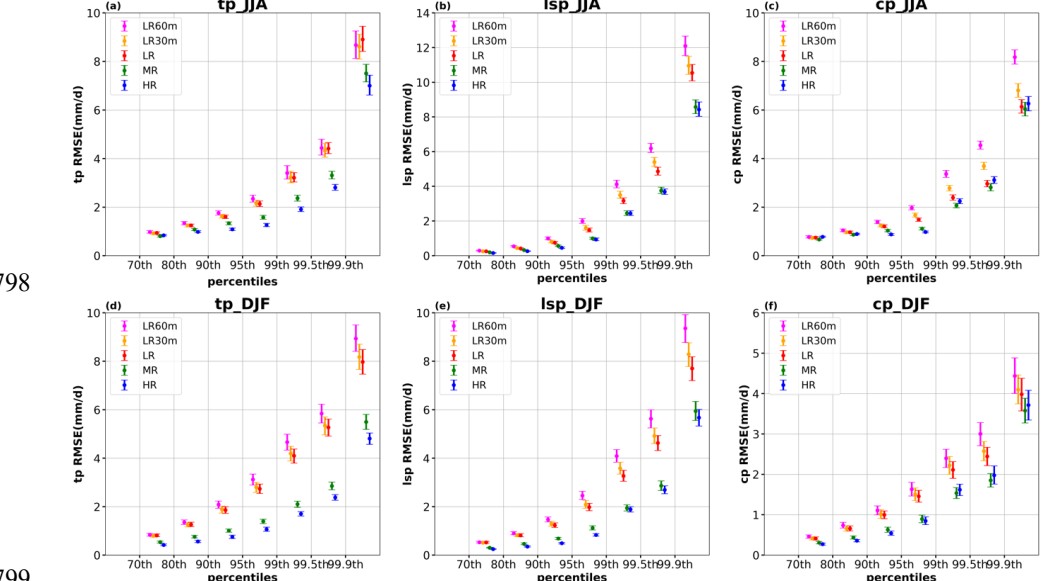


**Fig. 6** RMSEs of total precipitation (a & d) at different percentile ranges (70th – 80th, 80th –
90th, 90th – 95th, 95th – 99th, 99th – 99.5th, 99.5th – 99.9th and >99.9th) and the corresponding
large-scale precipitation (b & e) and convective precipitation (c & f) in OpenIFS simulations
(LR60m: magenta, LR30m: orange, LR: red, MR: green, HR: blue) against ERA5 over Europe
during 1982-2019. (a) – (c) are for JJA, and (d) – (f) for DJF. Dots are the RMSE values, and
error bars are the 95 % confidence intervals. Unit is mm/d.









**Table**
Table 1: The experiment details of different horizontal resolutions and model time steps in
OpenIFS.

|  | LR60m | LR30m | LR | MR | HR |
|---|---|---|---|---|---|
| Vertical resolution | L91 | | | L91 | L91 |
| Horizontal Resolution | 100 km (Tco95) | | | 50 km (Tco199) | 25 km (Tco399) |
| Time steps | 60 minutes | 30 minutes | 15 minutes | 15 minutes | 15 minutes |
