# Peer review of "Impact of horizontal resolution and model time step on European precipitation extremes in the OpenIFS 43r3 atmosphere model"

_Geoscientific Model Development, 2024_

## Author Comment (AC1)

**Response to reviewers**

Dear Editor and reviewers,

We appreciate your time and efforts in reviewing our manuscript and providing constructive comments. These comments are all insightful and helpful for improving our study. We have carefully considered the comments and updated the manuscript to address them. Below we provide point-by-point responses in blue. The line numbers refer to the tracked version of our manuscript.

**Response to Reviewer#1:**

**General comments**

The preprint authors address an important scientific question of theoretical and practical relevance: modelling the extreme part of precipitation with CMIP-style climate models. The introductory part follows a clear and logic structure by introducing the topic and its relevance, reviewing other model analysis studies in the field, recounting the origins of observational and reanalysis data used for comparison, introducing the details of the model simulations, and finally explaining the statistical terms and quantities. In particular the assessment of the strengths and weaknesses of the observational and reanalysis data sets allows the reader to understand the following analysis.

The article is well written in a clear language with distinct formulations, which allows to easily understand and follow the text.

**Specific comments**

The introduction is very helpful in introducing the relevant precipitation processes, related biases in CMIP models and (potential) links to spatial and temporal resolution. The detailed description of methods to compute quantiles highlights details that many readers may not be aware of. This is helps to understand and follow the analysis procedure in greater detail.

The analysis of results highlights improvement patterns (e.g. improvements over mountainous regions with higher horizontal resolution, better convection precipitation with shorter time steps), while at the same time not generalising where there is no evidence and also indicating when and why there might be no improvements (e.g. MR to HR). Overall, the conclusions appear convincing and detailed.

**Technical comments**

Two minor comments: Figures 2, 4 and 5 seem to be at the lower limit w.r.t. their size (when printed on A4 paper). Particularly the fonts in this figures is a bit hard to read.

**Thank you for pointing out these issues, we adjusted the size and fonts in these figures (see page 20-24).**

The text refers to figures S2, S3, S4, and S5 but the referee could not find these figures. However, it is possible that I am just not aware of some common practise in this journal. It is also noted that the code for those figures is included in the referenced Jupyter notebooks.

**Thank you and sorry to hear that you could not find supplementary figures S2, S3, S4 and S5. Actually, this is included in the additional material which you can see and find in the supplementary material at: https://gmd.copernicus.org/preprints/gmd-2024-66/**

**(You can find the supplementary materials on the top right corner, which has a red sign 'Download')**

**Response to Reviewer#2:**

**General comments**

This study focuses on the impact of horizontal resolution and model time step on extreme precipitation over Europe. The authors use AMIP-style simulations with the OpenIFS at various resolutions, covering a 25-year period (1979-2014). Results are compared with daily gridded precipitation observations data from GPCC and ERA5 reanalysis data. Although the study has its merits and is covering the very relevant topic of precipitation extremes in Europe, the approaches followed in evaluation and analysis of the results require substantial changes and reconsideration.

**Specific comments**

1. The general introduction of the topic this study addresses and the context and background information provided by previous studies is rather confusing and lacks focus. The authors cover a very wide range of studies, from CMIP-based studies (300-50km, multi-year runs) to even regional climate and seasonal studies at km-scale resolution. It can be improved if the authors focus their introduction in the temporal and spatial scales relevant for the current study, and discuss current limitation of the existing approaches and where their research fits into this context. Most of the information needed is already there and can be re-written in a more concise way.

   **Thank you for this helpful comment. We have rewritten the introduction (line 41-91).**

2. The novelty of the approach followed by the authors should be further highlighted. For example what is the difference of this study versus the one done by Strandberg & Lind (2021), which seems to cover exactly what the topic the authors addresses in this study, at even higher horizontal resolutions and with multiple models.

   **Thank you for this important suggestion. There are differences between this study and the one done by Strandberg & Lind (2021). Firstly, we analyzed the effect of both horizontal resolution and model time step on extreme precipitation using atmospheric model only, while Strandberg & Lind (2021) only focused on the horizontal resolution's impact on precipitation in coupled model simulations. Secondly, we also compared the large-scale and convective precipitation in different resolution and time step configurations, which was not studied in Strandberg & Lind (2021). Now we have added more information to make it more clear in line 82-89.**

3. The experimental setup could be further improved. What is the motivation behind using the same timestep for MR and HR experiment. I would expect the timestep for the 50km run to be 20 min. This would certainly have an impact on precipitation extreme with the MR experiments. Why not test the impact of model timestep across all model resolutions (i.e., including MR and HR experiments). This would make for a

rather interesting and novel study, where the authors could identify if timestep sensitivity for climate simulation of extreme precipitations changes across resolutions.

**Thank you and this is really a good point, I agree that it would be very interesting to identify if timestep sensitivity for simulation of extreme precipitation changes across resolutions. However, we usually do not use long time step in HR in OpenIFS, because it is not reasonable for some physical processes (e.g. advection) and we will get errors from semi-Lagrangian scheme. The time step for HR is usually 15 minutes or shorter. Different time steps are only tested for LR because the motivation of experimental setup is to investigate which time step at low resolution is suitable for coupled model simulations (Savita et al., 2024). Computational cost is also a limit that we can not test all time steps for high resolutions.**

**The reason we keep the same time step for LR, MR and HR experiment is that we wanted to investigate how the model precipitation will change due to changes in model horizontal resolution.**

4.  Regarding the model evaluation, although the use of the GPCC dataset is appropriate the use of ERA5 as reference benchmarks for evaluating RMSE of convective precipitation is not appropriate. My understanding is that ERA5 does not explicitly assimilate convective and large-scale precipitation rates, only total precipitation estimates. Also the authors show in Figures 1-3, that ERA5 is not substantially better (and not significantly different) at capturing precipitation at any percentiles compared to MR and HR simulations as RMSE for ERA5 lies often inside the CIs of MR and HR. Hence this does not justify using it as an evaluation benchmark for convective precipitation to compute RMSE scores. Although the discussion about difference in convective vs large-scale precipitation is interesting and could be including as a indication of differences (preferably showing them as PDFs all the different precipitation types) for each experiment, the current evaluation done in section 3.2 is not appropriate.

**Thank you for this valuable comment. ERA5 does not assimilate total precipitation, which is calculated from the combined large-scale and convective precipitation. Observation datasets such as GPCC and GPCP do not provide convective and large-scale precipitation separately. ERA5 is not the truth, the reason we took ERA5 as benchmarks is that ERA5 has smaller biases in simulating precipitation in extra-tropics (Lavers et al., 2022), we would expect the convective and large-scale precipitation also have smaller biases. In section 3.2, we do not focus much on how the model simulations are close to ERA5, we more care about whether the differences of each simulation and ERA5 (represented by RMSE) are significantly different to each other. That means we more focus on the error bar (Fig. 6) overlaps with others, but not their values. We also updated some texts (line 491-498, 518).**

**As you suggested, we added a frequency distribution plot of cp and lsp in the supplementary Fig. S6 & 7. We also added an additional paragraph to discuss the**

**physical processes involved (line 444-459). This also clarify the comment 5 you mentioned.**

5. During the analysis of the results in section 3.2 the authors often provide remarks about convective and large-scale precipitation differences between the different resolutions for summer and winter, without discussing the physical processes involved. For example it would help to clarify that convective precipitation is generally related with not explicitly resolved convective motions, and deep convection systems with scales smaller than the effective resolution of the model (e.g., Mediterranean Hurricanes or MCS) which tend to contribute more precipitation around the Mediterranean. On the other hand, large-scale precipitation is likely to originate from large-scale synoptic storms at these resolutions. As the effective resolution of the model increases the ratio between convective and large-scale precipitation will tend to change, since more convective motions are resolved rather than parametrized. Again plotting a frequency distribution for precipitation types for the different experiments will really help here.

    **Thank you very much and please see the response to comment 4.**

6. Line 53: What to the authors mean by "lack of observations"? Does this mean that there is no assimilation of precipitation observations in climate models, or that we lack observations of precipitation to built better parametrization schemes?

    **Thank you for pointing out this issue. We mean that we don't have long-term observations for precipitation which are very important for evaluating the climate models. We rewrote the sentences to make it clear (line 46-50).**

7. Line 133-136. What time of interpolation is used here to convert the data to the regular grid and also from the native resolution for MR and HR experiments to the intermediate resolution of 0.45 degrees. Is it similar to the one mentioned in lines 170-174?

    **Thank you for your comment. Bilinear method is used by XIOS to convert the data from native grid data to regular grid, also from native resolution for MR and HR to the intermediate resolution of 0.45 degrees. It is not the same as the one mentioned in lines 170-174, which is the second-conservative method. We redid one-year (1979) simulation using the HR model and saved the output on a native grid. We converted the native grid data to regular grid using bilinear and conservative method separately. We found that these two distributions are similar, the bilinear one has slightly more extreme events than the conservative one. The 99$^{th}$ percentile precipitation for the two distributions are very close (17.8 mm/d for bilinear and 17.4 mm/d for conservative method), therefore, we think the interpolation method would not have a large effect on the final conclusion.**

[Figure]

**Figure. Frequency distribution of European precipitation converted from native grid to regular grid by bilinear (blue) and conservative (red) method in 1979. The dash lines are the 99th percentile value for each distribution.**

8. Line 260-261: How is this sentence related to the overestimation of precipitation over the Alps in the LR experiments? Lavers et al. (2022) is about a single storm in ERA5, rather than the consistent multi-year estimations of 99th precipitation percentiles. Also ERA5 resolution is 31km, which is very different than the LR experiment and closer to the HR experiment, and based on Figure 2 I can't find a substantial overestimation of precipitation in the northern side of the Alps for ERA5 or HR.

**Thank you very much for your suggestion. We agree that both the time scales and resolutions are different between the single storm in ERA5 and consistent multi-year estimations of 99th precipitation percentiles in LR. It is not appropriate to compare them, therefore, we deleted that sentence.**

9. Line 271: Have authors checked the data from GPCP (instead of GPCC) to see if the high values near Slovenia are also present in that dataset. This may help with diagnosing the source of the bias in precipitation over that region.

**Thank you for this idea. We checked the data from both GPCP and EOBS, and found the similar negative bias near Slovenia in EOBS, but not in the GPCP. We have added this information in the manuscript (line 391-399) and a plot in supplementary Fig. S8.**

10. Line 433: I think Jung et al. (2012) does not discuss at all changes in tropical cyclone intensity, but only extratropical cyclone intensity.

**Thank you for pointing out this issue and it is fixed now (line 589-594).**

---

## Referee Report (RR1)

I thank the authors for providing a succinct article investigating the sensitivity of extreme precipitation events in OpenIFS to model time step and horizontal resolution. I have tried to make some minor edits to improve the readability of your article in places but in general the quality of scientific writing and accompanying figures is very good. I have one substantive point for you to think about in relation to RMSEs at different percentile thresholds – and specifically what precipitation amounts these percentiles correspond to. I also encourage the authors to consider using "finer (or coarser) horizontal resolution" and "shorter (or longer) model time step" as it will help the readability of your article.

Substantive point

In Fig. 5 you show the RMSE for different percentiles thresholds and state the "RMSEs increase exponentially with increasing percentiles". It would be useful here to have some additional knowledge of what rainfall amount the e.g. 70th, 80th and 90th percentiles correspond to. For example, the 80th percentile may correspond to only perhaps <2mm/day, in which case would it follow that the RMSE for all resolutions (and time step lengths) would be much smaller at these lower (less extreme) percentiles? Lines 285-288 – is RMSE the best statistic to use to support this conclusion? Given the discussion above around rainfall amounts that correspond to each percentile – I'd advise calculating and/or stating the rainfall amounts that the 70th, 80th and 90th percentiles correspond to.

Minor comments

In section 3.1 (**lines 206-210**) - I think it is worth starting that ERA-5 is a reanalysis product that assimilates observations of precipitation. Therefore we should expect it to e.g. do a better job of reproducing dry (1994) and wet (2010) years – than OpenIFS AMIP simulation that doesn't "see" the precipitation observations.

**Line 233** – missing word – add percentile after 99th

**Lines 240-242** – also in Fig. S1 and Fig. S2 there looks to be a shift northwards of the highest orography in LR compared with MR and HR – this could be brought into this discussion.

**Line 293** – suggest reversing statement to make it clearer to the reader what you mean – i.e. shorter timestep corresponds to smaller RMSE. e.g. "The RMSEs for LR, LR30m, and LR60m are smaller when the model time step is shorter. However…"

**Line 303** – I think it is more common to say "unresolved" convective motions in this context.

**Line 306** – suggest to help the reader follow what you are saying here "When moving to finer (or higher) horizontal resolutions, large-scale precipitation is likely to increase"

**Line 312** – mostly "consistent of" or shorten to "extreme precipitation is mostly large-scale…"

**Lines 317-319** – 40N also cuts out a significant part of southern Spain, Greece and far south of Italy and Sicily. This may be intended but I suggest its worth stating explicitly that you are cutting off more than just North Africa.

**Line 348** – Revise wording slightly for readability - "It is likely due to "the"(?) large-scale precipitation "increasing"(?) by a larger percentage…". Can you provide any evidence to support this statement? E.g. "This is due to the large-scale precipitation increasing by XX%, whereas convective precipitation only increases by YY%."

**Lines 345-353** – this section needs some editing and in places is a bit weak e.g. "it is likely" – try to be more concise and provide evidence to support these statements. As is stands I'm not sure this paragraph adds to the results section.

**Line 356** – consider saying a "shorter model time step" (and elsewhere in this paragraph).

I think the section detailing native grid results should be within its own sub-section under the results heading – to give clarity to the reader that you are re-producing earlier results but without the re-gridding step applied previously.

**Line 395** – remove the word "and" – use a comma after (Fig. 11 a-d) instead.

**Line 395** – change "coarsened resolution results" to "regridded results"

**Line 397 –** as above "regridded results". Coarsened resolution to the reader implies you are discussing the resolution of the model whereas my understanding is that here you are comparing with earlier results where you had regridded the higher-resolution data to a common 0.9 degree grid.

Results and discussion section – be careful in your use of "increasing horizontal resolution" – perhaps change to "finer horizontal resolution" or similar.

The final paragraph of your discussion on precipitation observational datasets is interesting. Given some of the issues that you correctly highlight I wonder whether it would be worth considering adding a line about future work that would follow-on from this study. You could compare model output to "an ensemble" of observational products – this would helpfully provide a spread of observational estimates and allow insights on whether and which model configurations sit within the observational spread.

Minor changes to improve figures

**Fig. 1** – consider removing vertical grid lines to reduce clutter on the plot. Update the figure line labels (e.g. capitalise RMSE).

**Fig. 3 and Fig. 4** – could you change the panel layout so that each panel is larger. I understand that you want to show differences to obs below relevant model data but its very challenging to see significance stippling at current image size

**Table 1**- In HR column – the "native" output resolution is not 400x800 – this data has already been coarsened or regridded as stated in lines 384-385 of the manuscript. For clarity suggest you state the native resolution and in brackets add the data resolution you used for your evaluation.

---

## Editor Decision (ED1)

Dear Authors,

Your paper has been submitted to a second round of reviews and there are still major concerns with your revised version. Therefore, before making my final decision on the publication or the rejection of your paper, I would like to have your feedback to the second round of reviews. I put here below the remarks made by the reviewers in the second round and I am asking you to reply point by point.

Reviewer 1:

This study focuses on the impact of horizontal resolution and model time step on extreme precipitation over Europe. The authors use AMIP-style simulations with the OpenIFS at various resolutions, covering a 25-year period (1979-2014). Results are compared with daily gridded precipitation observations data from GPCC and ERA5 reanalysis data. Although the authors did address some of the previous remarks/comments and adjusted parts in the revised manuscript, some of the original points raised have not been adequately addressed.

Specific comments:

1. The introduction is substantially improved and sharpened in the revised manuscript.

2. Lines 107-113: Although the authors responded with additional information to previous comment regarding the interpolation method used to convert the model output from native to regular grid, the manuscript has not been adapted to include this information. I would recommend that the authors adjust the manuscript to match their response to previous comments.

Regarding XIOS server output re-gridding, I could only find information in literature that the XIOS server is doing conservative (1st or 2nd) order interpolation, not bilinear interpolation. Could the authors please provide the relevant documentation on the interpolation methods of the XIOS server, either through appropriate referencing of documentation that includes the information on the interpolation methods or by making available the interpolation source code? That would substantially strengthen the reproducibility of the study.

3. I still have strong reservations regarding the use the ERA5 convective and large-scale precipitation as benchmark for comparing the HR, MR and LR convective and large-scale precipitation. Although Lavers et al. (2022) compares ERA5 total precipitation to observations and does indeed find that in Extratropics ERA5 it has lower biases than in the Tropics, there is no comparison in Lavers et al. (2022) against different model output simulations that would show that ERA5 is doing better than OpenIFS or any other model at similar resolution. Neither does Lavers et al. (2022) suggest in their manuscript that ERA5 large-scale precipitation (LSP) or convective precipitation (CP) can be used as proxies for observed large-scale or convective precipitation.

Since the authors own results in this study suggest that ERA5 is doing similarly or slightly (for precipitation percentiles higher than 99%) better than HR and MR experiments against observations, I would argue that assuming that LSP and CP precipitation from ERA5 is better than HR and MR experiments is not valid. Hence, I would recommend that the authors avoid

the use of a metric such as RMSE, which hints to proper evaluation against a reliable benchmark, to compares LSP and CP precipitation between ERA5 and LR, MR and HR experiments. Instead, the authors can use and compare the precipitation distributions for LSP and CP and discuss about differences between LSP and CP across the different resolutions or for the different percentiles (as done in Figs S6-S7), without suggesting that ERA5 convective and large-scale precipitation can be used as a valid "observations" (i.e., removing Fig 6 and RMSE comparison in Figs 4,5).

4. Given that the focus of the study is on extreme precipitation, which can be quite sensitive to resolution and tends to occur very locally (e.g., scales less than ~1x1 degree) have the authors investigated whether the conclusions of the study change if precipitation is investigated in native resolution rather than the coarsely interpolated at ~1x1 degrees? Also, if the interpolation from native grid to the regular grid is indeed bilinear instead of conservative as the authors suggest in their response, this would mean that total precipitation in no longer conserved when interpolating to ~1x1 degree in MR and HR experiments. Have the authors checked how this affects the conclusions of this study? The impact of interpolation method on extreme precipitation, and impact of coarsening from native resolution to ~1x1 degrees, should ideally be discussed in the manuscript.

5. Lines 296-298. Assuming directly that convective precipitation decreases with increasing resolution may not be valid for all resolutions. The ratio between convective and large-scale precipitation will change, likely because large-scale precipitation tends to increase with increasing resolution. But so can total precipitation. It would be more accurate if the authors mention that the ratio between convective and large-scale precipitation can change as resolution increases. Also, the origin of large-scale precipitation would depend on the model's effective resolution, hence it is not accurate to say that large-scale precipitation originates from synoptic storms without mentioning for which resolution this assumption may be valid.

6. Lines 432-434. What is the link here between the extreme precipitation in Europe and tropical cyclone representation? This is not investigated in the current study. Also, the study from Manganello et al. (2011) is rather old if the authors want to make a comparison with ECMWF-IFS system on tropical cyclone representation, as the current operational resolutions is 9 km.

Reviewer 2 :

The paper is scientifically sound and well written. If the following minor comments can be addressed, then I believe the paper is suitable for publication in GMD.

1) Introduction: The sensitivity of climate model performance to horizontal resolution and model timestep is presented as applying to all models. This is probably true to some extent, but the level of sensitivity varies considerably between models and this should be acknowledged. The IFS appearing to be amongst the more sensitive in this regard. I think it's also work stating in the Introduction that the purpose of parametrizations is to represent processes which are not resolved (spatially or temporally), hence this sensitivity reflects a weakness in the formulation of those models.

2) Line 75: I don't think Jung et al. particularly comment on timestep sensitivity of precipitation. They indicate that the errors in the tropical circulation are smaller at 15 min than 60 min timesteps (opposite of what's written here).

3) Line 76: The Roberts et al. (2018) reference is not listed in the final Bibliography.

4) Section 3: The authors have a number of figures as supplementary material, however the number of figures in the main article is not excessive and the manuscript is reliant on the figures in the supplementary material for the reader to follow (e.g. paragraph starting line 257). I therefore suggest most/all of the figures in the supplementary material are moved to the main paper.

5) Line 297: Given that few convective systems are larger than 25km, I would question whether convective precipitation should decrease with increased resolution over the range of resolutions studied here – this would imply >25km^2 updraughts from explicit representation. (I would expect the resolution sensitivity as one moved to higher resolutions with grid lengths <20km). Does this point to a weakness in the convective parametrization?

6) Paragraph starting line 311: It should be noted in the manuscript that comparing the split between convective and large-scale precipitation in ERA5 is also justified because it shares the same convective parametrization as the IFS. It would be less appropriate for other models for which the definition of convective precipitation is an arbitrary choice and related to the formulation of the convective parametrization and what it handles versus the large-scale scheme (convective core?, anvils?, etc.).

7) Line 403/404: The three references should be listed in the same order as the three example models to which they relate. I also wonder if lines 400-406 would be better in the Introduction?

8) Line 429: The need to re-tune for different resolutions appears to be presented as fact, whereas if model parametrizations were appropriately scale aware, tuning for different resolutions shouldn't be needed.

9) Figure 1: Colors/line styles of lines in the figure don't match what's written in the caption.

10) Map plots (e.g. Figure 2): These would be better as block plots (i.e. each grid box is filled with a color) rather than using a filled coloring which can be misleading e.g. if single grid boxes are notably different to surroundings.

11) Figures 4&5: Given that the vast majority of extreme precipitation over northern Europe is large scale, and over southern Europe is mostly large-scale in DJF, and it is the large-scale precipitation which shows the strongest resolution dependence of the RMSE, would it be better to plot Figures 4 and 5 as lsp/tp?

---

## Author Response (AR2)

**Point-to-point response to the Referees' comments**

**Impact of horizontal resolution and model time step on European precipitation extremes in the OpenIFS 43r3 atmosphere model**

Yingxue Liu[1,2], Joakim Kjellsson[1,2], Abhishek Savita[1] and Wonsun Park[3,4]

**Dear Editor and reviewers,**

**We appreciate your time in reviewing our manuscript and providing helpful feedback. We have carefully considered the comments and revised the manuscript accordingly. Below we provide point-by-point responses to the comments in blue. The line numbers refer to the tracked version of our manuscript.**

**Reviewer 1:**

This study focuses on the impact of horizontal resolution and model time step on extreme precipitation over Europe. The authors use AMIP-style simulations with the OpenIFS at various resolutions, covering a 25-year period (1979-2014). Results are compared with daily gridded precipitation observations data from GPCC and ERA5 reanalysis data. Although the authors did address some of the previous remarks/comments and adjusted parts in the revised manuscript, some of the original points raised have not been adequately addressed.

**Specific comments:**

1. The introduction is substantially improved and sharpened in the revised manuscript.

Thank you very much.

2. Lines 107-113: Although the authors responded with additional information to previous comment regarding the interpolation method used to convert the model output from native to regular grid, the manuscript has not been adapted to include this information. I would recommend that the authors adjust the manuscript to match their response to previous comments.

Regarding XIOS server output re-gridding, I could only find information in literature that the XIOS server is doing conservative (1st or 2nd) order interpolation, not bilinear interpolation. Could the authors please provide the relevant documentation on the interpolation methods of the XIOS server, either through appropriate referencing of documentation that includes the information on the interpolation methods or by making available the interpolation source code? That would substantially strengthen the reproducibility of the study.

We are sorry for not providing a complete and detailed response in the first round of review. Below we are providing all the information you asked.

We checked with XIOS development team, and they confirm that XIOS uses the 1st or 2nd (default) order conservative method for interpolation. We did not define order attribute in our xml files which means XIOS uses the default 2nd conservative method to interpolate the model output from reduced Gaussian grid to regular grid (192x384 for LR, 400x800 for MR and HR). More details about the 2nd order conservative method can be found in Kritsikis et al. (2017). We also added this information in the manuscript (line 136).

3. I still have strong reservations regarding the use the ERA5 convective and large-scale precipitation as benchmark for comparing the HR, MR and LR convective and large-scale precipitation. Although Lavers et al. (2022) compares ERA5 total precipitation to observations and does indeed find that in Extratropics ERA5 it has lower biases than in the Tropics, there is no comparison in Lavers et al. (2022) against different model output simulations that would show that ERA5 is doing better than OpenIFS or any other model at similar resolution. Neither does Lavers et al. (2022) suggest in their manuscript that ERA5 large-scale precipitation (LSP) or convective precipitation (CP) can be used as proxies for observed large- scale or convective precipitation.

Since the authors own results in this study suggest that ERA5 is doing similarly or slightly (for precipitation percentiles higher than 99%) better than HR and MR experiments against observations, I would argue that assuming that LSP and CP precipitation from ERA5 is better than HR and MR experiments is not valid. Hence, I would recommend that the authors avoid the use of a metric such as RMSE, which hints to proper evaluation against a reliable benchmark, to compares LSP and CP precipitation between ERA5 and LR, MR and HR experiments. Instead, the authors can use and compare the precipitation distributions for LSP and CP and discuss about differences between LSP and CP across the different resolutions or for the different percentiles (as done in Figs S6-S7), without suggesting that ERA5 convective and large-scale precipitation can be used as a valid "observations" (i.e., removing Fig 6 and RMSE comparison in Figs 4,5).

Thank you for your advice. As you suggested, we have now removed the RMSE comparison between ERA5 and model simulated cp and lsp which were previously shown in Fig 4 & 5, and Fig 6. Now we discuss the distribution of cp and lsp across different horizontal resolutions and model time steps including the ratio between cp and lsp, which you can find in section 3.2 (line 337-443).

4. Given that the focus of the study is on extreme precipitation, which can be quite sensitive to resolution and tends to occur very locally (e.g., scales less than ~1x1 degree) have the authors investigated whether the conclusions of the study change if precipitation is investigated in native resolution rather than the coarsely interpolated at ~1x1 degrees? Also, if the interpolation from native grid to the regular grid is indeed bilinear instead of conservative as the authors suggest in their response, this would mean that total precipitation in no longer conserved when interpolating to ~1x1 degree in MR and HR experiments. Have the authors checked how this affects the conclusions of this study? The impact of

interpolation method on extreme precipitation, and impact of coarsening from native resolution to ~1x1 degrees, should ideally be discussed in the manuscript.

Thank you for your advice. As we responded to comment 2, that is, the second-order conservative method is used for interpolation, it means that total precipitation is conserved when interpolating from model native grid to regular grid. Therefore, we only discuss the impact of coarsening from native resolution to 1x1 degree here. The native resolutions of LR, MR and HR are 100 km, 50 km and 25 km, respectively. However, we saved the model output as 192x384 for LR, and 400 x800 for MR and HR.

We analysed the European extreme precipitation, and distribution of cp and lsp on native resolution and we have seen small changes in results between native and coarsened resolution (interpolated to 1x1 degree), and it is discussed in detail in the manuscript (line 445-465).

5. Lines 296-298. Assuming directly that convective precipitation decreases with increasing resolution may not be valid for all resolutions. The ratio between convective and large-scale precipitation will change, likely because large-scale precipitation tends to increase with increasing resolution. But so can total precipitation. It would be more accurate if the authors mention that the ratio between convective and large-scale precipitation can change as resolution increases. Also, the origin of large-scale precipitation would depend on the model's effective resolution, hence it is not accurate to say that large-scale precipitation originates from synoptic storms without mentioning for which resolution this assumption may be valid.

Thank you very much, and we have included your suggestions and modified the text accordingly in the manuscript (line 339-345).

6. Lines 432-434. What is the link here between the extreme precipitation in Europe and tropical cyclone representation? This is not investigated in the current study. Also, the study from Manganello et al. (2011) is rather old if the authors want to make a comparison with ECMWF-IFS system on tropical cyclone representation, as the current operational resolutions is 9 km.

Thank you, here we want to say that we found a large improvement from LR to MR, but not much improvement from MR to HR. This diminishing return is valid for European extreme precipitation in this study, also for climatological variables in other studies, but may not be valid for tropical extreme precipitation. Because tropical cyclones, which often cause tropical extreme precipitation, are better simulated at 16 km, but not at 126 and 39 km. We make some changes for this paragraph in the manuscript (line 642-652).

**Reviewer 2 :**

The paper is scientifically sound and well written. If the following minor comments can be addressed, then I believe the paper is suitable for publication in GMD.

1) Introduction: The sensitivity of climate model performance to horizontal resolution and model timestep is presented as applying to all models. This is probably true to some extent, but the level of sensitivity varies considerably between models and this should be acknowledged. The IFS appearing to be amongst the more sensitive in this regard. I think it's also work stating in the Introduction that the purpose of parametrizations is to represent processes which are not resolved (spatially or temporally), hence this sensitivity reflects a weakness in the formulation of those models.

Thank you for the suggestions. We now added the suggested information to manuscript in the introduction section (line 91-95).

2) Line 75: I don't think Jung et al. particularly comment on timestep sensitivity of precipitation. They indicate that the errors in the tropical circulation are smaller at 15 min than 60 min timesteps (opposite of what's written here).

Thank you for pointing this out. We fix it in the manuscript now (line 85-87).

3) Line 76: The Roberts et al. (2018) reference is not listed in the final Bibliography.

 It is fixed now (line 897-900).

4) Section 3: The authors have a number of figures as supplementary material, however the number of figures in the main article is not excessive and the manuscript is reliant on the figures in the supplementary material for the reader to follow (e.g. paragraph starting line 257). I therefore suggest most/all of the figures in the supplementary material are moved to the main paper.

Thank you for your suggestion. We moved a few figures to the main paper (Fig. 3 & 4).

5) Line 297: Given that few convective systems are larger than 25km, I would question whether convective precipitation should decrease with increased resolution over the range of resolutions studied here – this would imply >25km^2 updraughts from explicit representation. (I would expect the resolution sensitivity as one moved to higher resolutions with grid lengths <20km). Does this point to a weakness in the convective parametrization?

In this study, we use 'large-scale precipitation' to represent precipitation resulting from resolved processes whose scales are larger than the model resolution, and 'convective precipitation' to represent the precipitation from processes smaller than the model resolution which is need to be parameterized. That is, the convective precipitation in OpenIFS is not only from convective systems smaller than 25 km, but also from physical processes that are not resolved by OpenIFS at its resolution (i.e., <100 km for LR, <50 km for MR and <25 km for HR). The convective precipitation may decrease with increased resolution is likely because higher resolution can resolve more small-scale processes than lower resolution, thus less processes

are needed to be parameterized (Hertwig et al., 2015). In the manuscript, it is not accurate to assume that convective precipitation decreases with increased resolution without mentioning according resolution, therefore, we made some changes in the manuscript by mentioning lsp and the ratio between cp and lsp (line 339-345).

6) Paragraph starting line 311: It should be noted in the manuscript that comparing the split between convective and large-scale precipitation in ERA5 is also justified because it shares the same convective parametrization as the IFS. It would be less appropriate for other models for which the definition of convective precipitation is an arbitrary choice and related to the formulation of the convective parametrization and what it handles versus the large- scale scheme (convective core?, anvils?, etc.).

Thank you very much and this is a good point, we agree that it is more appropriate to compare cp and lsp in OpenIFS and ERA5, as OpenIFS and IFS have the same convective parametrization. Although the comparison is appropriate, using ERA5 as benchmark is not very reasonable as we cannot prove ERA5 is doing better than OpenIFS as reviewer 1 suggested. Therefore, to avoid using ERA5 as proxies for observed data, we remove the RMSE comparison of cp and lsp between ERA5 and OpenIFS. Instead, we discuss the distribution of cp, lsp and their ratio across horizontal resolutions and model time steps in section 3.2 (line 337-443).

7) Line 403/404: The three references should be listed in the same order as the three example models to which they relate. I also wonder if lines 400-406 would be better in the Introduction?

Thank you. We adjusted the order of references to the example models as you suggested (line 483-484), and edited 400-406 to make it clear (line 480-484).

8) Line 429: The need to re-tune for different resolutions appears to be presented as fact, whereas if model parametrizations were appropriately scale aware, tuning for different resolutions shouldn't be needed.

Thank you very much, as that sentence is a bit confusing, therefore, I removed that sentence.

9) Figure 1: Colors/line styles of lines in the figure don't match what's written in the caption.

Thank you for pointing out that typo, it is fixed now (line 960-961, Fig. 1)

10) Map plots (e.g. Figure 2): These would be better as block plots (i.e. each grid box is filled with a color) rather than using a filled coloring which can be misleading e.g. if single grid boxes are notably different to surroundings.

Thank you, we replace the map plots with block plots (Fig. 2 & 3 & 4).

11) Figures 4&5: Given that the vast majority of extreme precipitation over northern Europe is large scale, and over southern Europe is mostly large-scale in DJF, and it is the large-scale precipitation which shows the strongest resolution dependence of the RMSE, would it be better to plot Figures 4 and 5 as lsp/tp?

Thank you for your helpful suggestions. We plot lsp/tp instead of cp/tp (now Fig. 6), and make corresponding changes in the text (line 346-351).

---

## Author Response (AR3)

**Point-to-point responses to Referee's comments**

**Impact of horizontal resolution and model time step on European precipitation extremes in the OpenIFS 43r3 atmosphere model**

Yingxue Liu[1,2], Joakim Kjellsson[1,2], Abhishek Savita[1] and Wonsun Park[3,4]

**Reviewer #1**

This study focuses on the impact of horizontal resolution and model time step on extreme precipitation over Europe. The authors use AMIP-style simulations with the OpenIFS at various resolutions, covering a 25-year period (1979-2014). Results are compared with daily gridded precipitation observations data from GPCC and ERA5 reanalysis data. The authors did efforts to address some of the previous remarks/comments, adjusted parts in the revised manuscript, and provide clarification regarding the interpolation methods and impact on the results. Although I do appreciate the effort, I still have a major concern regarding the analysis conducted in section 3.2 and Figures (7,8, 11 and S5).

Specific comments:

The analysis in section 3.2 is based largely on Figures 7,8,11 and S5, S6. However, looking closer at these figures I can see that the distributions include probability densities for negative values of precipitation (both lsp and cp) as well as their ratio. To my understanding it should not be possible to have negative daily precipitation rates.

I would guess that this could be something as simple as small negative values for precipitation, which can occur due to interpolation issues. If that is the case then the analysis in section 3.2 could remain largely valid (for precipitation > 0 mm). If the issue is something else, like adding precipitation difference fields (e.g., HR - ERA5) in the computation of PDFs, instead of the actual precipitation fields, then they analysis in section 3.2 may not be valid and would need to be corrected.

Could the authors please elaborate on what has caused the issue with these Figures and if this affects the analysis in section 3.2.

Thank you for your comments. The distribution figures (Figures 7, 8, 11 and S6, S7) were done by 'seaborn.kdeplot'. Because the smoothing algorithm of 'seaborn.kdeplot' uses a Gaussian kernel, the estimated density curve can extend to values that do not make sense for a particular dataset. For example, the curve may be drawn over negative values when smoothing data that are naturally positive (https://seaborn.pydata.org/generated/seaborn.kdeplot.html). We now use parameters to truncate the curve at the data limit, so that only the exact data we have are shown in the figures. All the distributions figures are updated (Figures 7, 8, 11 and S6, S7). It does not affect the analysis in section 3.2.

Minor comments:

I would suggest to relocated the paragraph in lines 382-392 of the revised manuscript to either section 3.1 and the discuss. The insights provided here are very valuable as they highlight the impact of the interpolation from native grid to the one used for the analysis and are somewhat misplaced here in section 3.2 which focuses on relative contributions for large-scale and convective precipitation.

Thank you for your suggestions and we have moved the analysis on native grid to the discussion section (line 540-560).

Also, it may be interesting (although maybe not necessary or could be added as supplementary material) to compute the RMSE against CPCC also in the native resolution, just to see if the larger differences see in native resolution Fig 9, vs Fig 1 have any impact on the RMSE for the different percentiles.

Thank you for your advice. We cannot compute RMSE against GPCC in the native resolution for different percentiles (like we did in Fig. 5) without interpolation, because LR (192x384), MR and HR (400x800) have different resolutions with GPCC (180x360). For the 99th precipitation time series shown in Fig. 1 and Fig. 9, we can compute RMSE, because the curves have the same time dimension. We have computed the RMSE against GPCC for the time series curves in both regridded and native resolution for different percentiles (Fig. S8), and found larger RMSE values across different percentiles and larger sensitivity to horizontal resolution on native resolution (line 550-553).

I would also recommend to update the zenodo repository for this study to include the scripts used for plotting the new Figures added to this manuscript.

Thank you and we have updated the scripts on the zenodo repository (line 597).

**Reviewer #2**

I thank the authors for providing a succinct article investigating the sensitivity of extreme precipitation events in OpenIFS to model time step and horizontal resolution. I have tried to make some minor edits to improve the readability of your article in places but in general the quality of scientific writing and accompanying figures is very good. I have one substantive point for you to think about in relation to RMSEs at different percentile thresholds – and specifically what precipitation amounts these percentiles correspond to. I also encourage the authors to consider using "finer (or coarser) horizontal resolution" and "shorter (or longer) model time step" as it will help the readability of your article.

**Substantive point**

In Fig. 5 you show the RMSE for different percentiles thresholds and state the "RMSEs increase exponentially with increasing percentiles". It would be useful here to have some additional knowledge of what rainfall amount the e.g. $70^{th}$, $80^{th}$ and $90^{th}$ percentiles correspond to. For example, the $80^{th}$ percentile may correspond to only perhaps <2mm/day, in which case would it follow that the RMSE for all resolutions (and time step lengths) would be much smaller at these lower (less extreme) percentiles? Lines 285-288 – is RMSE the best statistic to use to support this conclusion? Given the discussion above around rainfall amounts that correspond to each percentile – I'd advise calculating and/or stating the rainfall amounts that the $70^{th}$, $80^{th}$ and $90^{th}$ percentiles correspond to.

Thank you for your advice. We added the precipitation amounts at different percentiles to Fig. 5, and discussed the relative RMSE (RRMSE, i.e., RMSE/ (GPCC precipitation)) in the manuscript (line 321-334). Differ from the smaller RMSEs at lower percentiles, the RRMSEs are comparable at lower (70-90$^{th}$) and higher percentiles (above 99$^{th}$).

For the lines 285-288 (now line 300-304), we concluded that extreme precipitation is more sensitive to horizontal resolution than lower percentile precipitation. This conclusion is based on RMSE which measures the absolute magnitude of biases, but not totally applicable on RRMSE. For RRMSE, a slightly larger sensitivity to horizontal resolution at higher percentiles holds from 90$^{th}$ (~6 mm/d) to >99.9$^{th}$ (~39 mm/d), however, precipitation's sensitivity to horizontal resolution at 70-90$^{th}$ percentile is comparable to that above 99$^{th}$ percentile. We have added a paragraph about RRMSE in the manuscript (line 321-334).

**Minor comments**

In section 3.1 (lines 206-210) - I think it is worth starting that ERA-5 is a reanalysis product that assimilates observations of precipitation. Therefore we should expect it to e.g. do a better job of reproducing dry (1994) and wet (2010) years – than OpenIFS AMIP simulation that doesn't "see" the precipitation observations.

Thank you and we have updated this information in the manuscript (line 214-217).

Line 233 – missing word – add percentile after 99$^{th}$

It is fixed now (line 244).

Lines 240-242 – also in Fig. S1 and Fig. S2 there looks to be a shift northwards of the highest

orography in LR compared with MR and HR – this could be brought into this discussion.

Thank you and it is a good point. The previous Fig. S2 & S3 are for (0 – 40° E, 62° N) which is in Scandinavia. We add the surface height over Alps region (~12.6° E, 45-50° N) in current Fig. S2, and found a lower and gentle mountain in LR than MR (HR) which can help to explain the precipitation bias over Alps (line 254-258).

Line 293 – suggest reversing statement to make it clearer to the reader what you mean – i.e. shorter timestep corresponds to smaller RMSE. e.g. "The RMSEs for LR, LR30m, and LR60m are smaller when the model time step is shorter. However..."

It is fixed now (line 308).

Line 303 – I think it is more common to say "unresolved" convective motions in this context.

It is fixed now (line 339).

Line 306 – suggest to help the reader follow what you are saying here "When moving to finer (or higher) horizontal resolutions, large-scale precipitation is likely to increase"

It is fixed now (line 342).

Line 312 – mostly "consistent of" or shorten to "extreme precipitation is mostly large-scale..."

It is fixed now (line 348).

Lines 317-319 – 40N also cuts out a significant part of southern Spain, Greece and far south of Italy and Sicily. This may be intended but I suggest its worth stating explicitly that you are cutting off more than just North Africa.

It is fixed now (line 356-357).

Line 348 – Revise wording slightly for readability - "It is likely due to "the"(?) large-scale precipitation "increasing"(?) by a larger percentage...". Can you provide any evidence to support this statement? E.g. "This is due to the large-scale precipitation increasing by XX%, whereas convective precipitation only increases by YY%."

It is fixed now (389-391).

Lines 345-353 – this section needs some editing and in places is a bit weak e.g. "it is likely" – try to be more concise and provide evidence to support these statements. As is stands I'm not sure this paragraph adds to the results section.

Thank you. We have removed some information and made it concise (line 384-395).

Line 356 – consider saying a "shorter model time step" (and elsewhere in this paragraph).

I think the section detailing native grid results should be within its own sub-section under the results heading – to give clarity to the reader that you are re-producing earlier results but without the re-gridding step applied previously.

Thank you and we have changed to 'shorter model time step' in the manuscript. We have also moved the analysis about native resolution to the discussion section (line 540-560).

Line 395 – remove the word "and" – use a comma after (Fig. 11 a-d) instead. Line 395 – change "coarsened resolution results" to "regridded results"

It is fixed now (line 554-555).

Line 397 – as above "regridded results". Coarsened resolution to the reader implies you are discussing the resolution of the model whereas my understanding is that here you are comparing with earlier results where you had regridded the higher-resolution data to a common 0.9 degree grid.

Thank you and we have fixed it in the manuscript.

Results and discussion section – be careful in your use of "increasing horizontal resolution" – perhaps change to "finer horizontal resolution" or similar.

Thank you and we have changed 'increasing horizontal resolution' to 'higher horizontal resolution' in the manuscript.

The final paragraph of your discussion on precipitation observational datasets is interesting. Given some of the issues that you correctly highlight I wonder whether it would be worth considering adding a line about future work that would follow-on from this study. You could compare model output to "an ensemble" of observational products – this would helpfully provide a spread of observational estimates and allow insights on whether and which model configurations sit within the observational spread.

Thank you and this is a good point. We have added this information that paragraph (line 579-583).

**Minor changes to improve figures**

Fig. 1 – consider removing vertical grid lines to reduce clutter on the plot. Update the figure line labels (e.g. capitalise RMSE).

It is fixed now (Fig. 1).

Fig. 3 and Fig. 4 – could you change the panel layout so that each panel is larger. I understand that you want to show differences to obs below relevant model data but its very challenging to see significance stippling at current image size

Fig. 3&4 have been updated.

Table 1- In HR column – the "native" output resolution is not 400x800 – this data has already been coarsened or regridded as stated in lines 384-385 of the manuscript. For clarity suggest you state the native resolution and in brackets add the data resolution you used for your evaluation.

It is fixed now (Table 1).

---

## Author Response (AR4)

**Point-to-point responses to Referee's comments**

**Impact of horizontal resolution and model time step on European precipitation extremes in the OpenIFS 43r3 atmosphere model**

Yingxue Liu[1,2], Joakim Kjellsson[1,2], Abhishek Savita[1] and Wonsun Park[3,4]

Dear Editor,

Thank you very much for your time and feedback on our manuscript. We have revised the the manuscript as you suggested. Below, we provide a point-by-point response to each comment.

Thank you again for your support.

l.376 and l.437: I think it is better to put a semicolon before the word "however"; i.e. write "... shorter model time step; however, convective precipitation ..." and "... and 800x1600 for HR; however, only 400x800 for HR ..."

Thank you and they are fixed in the manuscript (line 376 and line 437).

- l.435: add "it" before "may" in "Since the analysis of horizontal resolution's impact is based on regridded data, may therefore ...'

Thank you and it is fixed in the manuscript (line 435).